# Individually Harmful, Jointly Beneficial: Compression Strategy Interactions in Brain Tumor Segmentation

**Jiawei Chen**[1]
**Qizhe Zhang**[1]
**Hongsheng Chang**[1]
**Yuzhou Chen**[1]
**Guoxi Zhang**[1]
**Naohao Huang**[1]
**Shanghang Zhang**[1]
**Xueqing Yan**[1]                                    X.YAN@PKU.EDU.CN
[1] *Peking University, Beijing, China*

## Abstract

A counter-intuitive finding is reported in model compression for 3D brain tumor segmentation: two strategies that individually degrade or barely affect performance, Mamba block relocation and $\ell_2$-norm token pruning, recover near-baseline accuracy when combined under 2.2× parameter reduction. In experiments on SegMamba-V2 (BraTS 2023), Mamba relocation alone decreases Mean Dice by 2.45 percentage points at channel width $C=32$, yet the addition of token pruning reverses this loss entirely, achieving 91.23% Mean Dice (vs. 91.31% uncompressed baseline with 60.69M vs. 138.78M parameters). This positive interaction (+2.92 Dice points beyond additive expectation) is observed only at reduced capacity ($C=32$) and is absent at $C=48$ (−0.32), suggesting a state capacity bottleneck in which token pruning compensates for Mamba's difficulty processing high-resolution background tokens under limited channel width. The interaction is strongest for the Enhancing Tumor class (+3.91 points), the smallest and most boundary-sensitive target.

**Keywords:** model compression, brain tumor segmentation, interaction effects

## 1. Introduction

Three-dimensional medical image segmentation models based on State Space Models (Gu and Dao, 2024), such as SegMamba-V2 (Xing et al., 2026), achieve strong accuracy but require over 100M parameters. Model compression techniques, architectural modifications, token pruning (Bolya et al., 2023), channel width reduction, are commonly applied but typically evaluated in isolation. When combined, their joint effect may differ from the sum of individual effects, a phenomenon known as *interaction effects* in factorial experimental design (Montgomery, 2017). Recent work has begun to demonstrate that compression techniques are not orthogonal (Kim et al., 2026), yet systematic factorial analysis of such interactions remains rare (Isensee et al., 2021).

A complete 2×2×2 factorial experiment is conducted on SegMamba-V2 with BraTS 2023 (Menze et al., 2015; Bakas et al., 2017; Fathi Kazerooni et al., 2024), examining Mamba block relocation, $\ell_2$-norm token pruning, and channel width reduction ($C=48$ vs. $C=32$). Two individually ineffective or harmful strategies are found to produce a strong positive interaction at reduced capacity, recovering Mean Dice from 88.45% to 91.23% (near the uncompressed baseline of 91.31%).

## 2. Methods and Results

**Base model.** The base architecture is SegMamba-V2 (Xing et al., 2026) (Figure 1a). In the original design, TriMamba blocks reside in the low-resolution encoder stages (stages 2–3), while the high-resolution stages (stages 0–1) use only convolutional blocks.

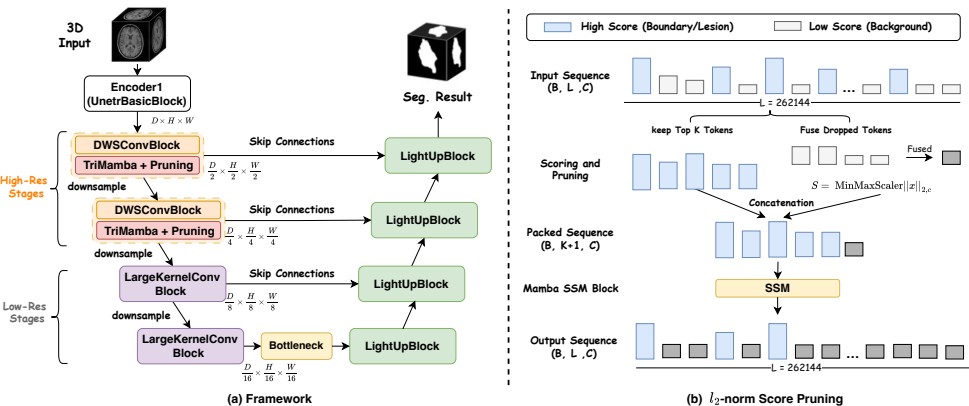

Figure 1: (a) SegMamba-V2 encoder: high-res stages use DWSConvBlock + TriMamba with optional pruning, low-res stages use LargeKernelConv. (b) $\ell_2$-norm pruning scores each token by its channel-wise $\ell_2$ norm (with Min–Max normalization). The top-$K$ tokens are kept, and the rest are fused into one summary token.

**Compression strategies.** Three binary factors are examined: (1) **Mamba block relocation**: the TriMamba blocks are moved from the original low-resolution stages (stages 2–3) to the high-resolution stages (stages 0–1), as shown in Figure 1a, so Mamba now processes high-resolution feature maps with substantially more spatial tokens. (2) $\ell_2$**-norm token pruning**: at each high-res stage, every spatial token $x \in \mathbb{R}^C$ is assigned an importance score equal to its channel-wise $\ell_2$ norm followed by Min–Max normalization, (3) **Channel width reduction**: from $C=48$ (138.78M params, 3723G FLOPs) to $C=32$ (61.71M params, 1661G FLOPs), a 2.2× reduction.

**Experimental setup.** All $2^3=8$ configurations are trained with fixed hyperparameters (SGD, polynomial LR with initial rate 0.01, decay $10^{-5}$, batch size 2300 epochs). Dice and HD95 for Whole Tumor (WT), Tumor Core (TC), and Enhancing Tumor (ET) are reported on the test set. Interaction effects are quantified as $I = y_{AB} - (y_A + y_B - y_\emptyset)$. The baseline ($C=48$) achieves Mean Dice 91.31%, consistent with 91.60% reported by Xing et al. (2026). Full details are in Appendix A.

**Results.** Three key patterns emerge from Table 1: *(1) Mamba relocation alone degrades performance* at both widths, most severely for ET ($-3.52\%$ at $C=32$). *(2) token pruning alone has minimal impact* ($<0.5\%$ Mean Dice change). *(3) a strong positive interaction emerges at $C=32$ but not $C=48$.* At $C=32$, the combination achieves Mean Dice 91.23%, exceeding the additive prediction (88.31%) by $+2.92$ points (HD95 interaction: $-9.79$.

Table 1: 2×2×2 factorial Dice (%) on BraTS 2023. **Bold**: best at $C$=32. $I$: Relocate×Prune interaction. HD95 in Appendix C.

| Relocate | Prune | $C$ | WT ↑ | TC ↑ | ET ↑ | Mean ↑ |
|---|---|---|---|---|---|---|
| ✗ | ✗ | 48 | 93.81 | 91.68 | 88.43 | 91.31 |
| ✗ | ✓ | 48 | 93.52 | 90.58 | 88.56 | 90.89 |
| ✓ | ✗ | 48 | 92.10 | 89.99 | 86.10 | 89.40 |
| ✓ | ✓ | 48 | 91.91 | 88.88 | 85.20 | 88.66 |
| | $I$ ($C$=48) | | +0.10 | −0.01 | −1.03 | −0.32 |
| ✗ | ✗ | 32 | 93.54 | 91.41 | 87.75 | 90.90 |
| ✗ | ✓ | 32 | 93.50 | 90.80 | 87.99 | 90.76 |
| ✓ | ✗ | 32 | 91.85 | 89.28 | 84.23 | 88.45 |
| ✓ | ✓ | 32 | **93.80** | **91.50** | **88.38** | **91.23** |
| | $I$ ($C$=32) | | +1.99 | +2.83 | +3.91 | +2.92 |

Appendix C). At $C$=48, the interaction is only −0.32. Per-class, ET drives the interaction most strongly (+3.91 at $C$=32), followed by TC (+2.83) and WT (+1.99).

**Analysis.** A hypothesis for this capacity-dependent pattern is proposed (detailed in Appendix D). In the original SegMamba-V2, Mamba operates at low-resolution stages where the token count is small and features are semantically rich. After relocation to high-resolution stages, Mamba must process far more spatial tokens, many from uninformative background. Mamba compresses sequential information into a fixed-size hidden state (Gu and Dao, 2024). At $C$=48, the 48-channel tokens provide enough capacity to tolerate this token flood, but at $C$=32, the state is overwhelmed.Token pruning resolves this by reducing the token count to foreground-relevant ones before Mamba processing (Jin et al., 2022).

## 3. Conclusion, Limitations and Future Work

Token pruning appears useless alone, yet becomes essential when combined with Mamba relocation at reduced capacity, consistent with recent evidence that compression strategies are not orthogonal (Kim et al., 2026).

**Limitations and self-critique.** Several important caveats constrain these conclusions. (1) All experiments are single training runs. Multi-seed tests (Appendix F) reveal substantial initialization sensitivity, which is the most significant threat. (2) Only one architecture (SegMamba-V2) and one dataset (BraTS 2023) are studied. Generalizability to other SSM-based models (Xing et al., 2024; Ruan and Xiang, 2024) is unknown. (3) The capacity-bottleneck hypothesis (Appendix D) is post-hoc and lacks mechanistic controls such as random token selection. (4) Only three compression dimensions are covered. Decoder lightweighting, scanning patterns, and alternative dropping criteria remain unexplored. (5) Only two capacity levels are examined. The interaction-capacity relationship may be non-monotonic. Future directions are discussed in Appendix J.

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

## Appendix A. Experimental Details

### A.1. Dataset and Preprocessing

BraTS 2023-GLI (Menze et al., 2015; Bakas et al., 2017; Fathi Kazerooni et al., 2024) comprises 1251 multi-modal MRI cases (T1, T1ce, T2, FLAIR) at approximately 1 mm isotropic resolution. The data are split into training/validation/test sets at a 70%/10%/20% ratio, yielding a fixed test set of 250 cases. Segmentation targets are Whole Tumor (WT), Tumor Core (TC), and Enhancing Tumor (ET).

Foreground volume statistics across the full dataset ($N$=1251) are summarized in Table 2. The mean foreground ratio is only 2.94%, confirming the extreme class imbalance typical of brain tumor segmentation. Edema (ED) constitutes approximately 63% of the foreground volume, while ET is the smallest component and therefore most sensitive to boundary delineation errors.

Table 2: Foreground volume statistics on BraTS 2023-GLI ($N$=1251).

|  | Mean | Std | Min | P25 | P50 | P75 | Max |
|---|---|---|---|---|---|---|---|
| Foreground ratio (%) | 2.94 | 1.73 | 0.09 | 1.54 | 2.75 | 4.15 | 10.34 |

Table 3: Per-class voxel count statistics on BraTS 2023-GLI ($N$=1251).

| Class | Mean | Median | Max | Zero-count |
|---|---|---|---|---|
| NCR (necrotic core, label 1) | 14,306 | 7,370 | 189,152 | 43 |
| ED (edema, label 2) | 60,215 | 52,314 | 216,411 | 1 |
| ET (enhancing tumor, label 3) | 21,447 | 17,337 | 111,250 | 33 |

### A.2. Training Configuration

All models are trained with fixed hyperparameters as listed in Table 4. Data augmentation follows the `light_training` strategy from the SegMamba-V2 codebase. Models with the Foreground Confidence Estimator (FCE) module additionally use a binary cross-entropy (BCE) auxiliary loss.

### A.3. Inference and Evaluation Protocol

Inference uses a sliding-window strategy with an ROI size of $128^3$ voxels and an overlap ratio of 0.5. Eight-fold mirror test-time augmentation (TTA) is applied along the three

Table 4: Training hyperparameters used across all configurations.

| Parameter | Value |
|---|---|
| Optimizer | SGD (momentum 0.99, Nesterov) |
| Initial learning rate | 0.01 |
| LR schedule | Polynomial decay |
| Weight decay | $1\times10^{-5}$ |
| Batch size | 2 |
| Epochs | 300 |
| Loss function | Cross-Entropy (+ BCE for FCE) |
| Data augmentation | `light_training` |
| Hardware | NVIDIA A100 80 GB |

spatial axes. Dice coefficients and 95th-percentile Hausdorff distance (HD95, in mm) are computed per class using the evaluation code from the SegMamba-V2 repository. The baseline reproduction ($C$=48, no modifications) achieves Mean Dice 91.31%, compared to 91.60% reported by Xing et al. (2026).

### A.4. Token Pruning Configuration

For the $\ell_2$-norm token pruning factor, the per-stage keep rates are set to $[K_0, K_1] = [0.5, 0.7]$, where $K_0$ applies to the first Mamba stage (Stage 2, sequence length 4096) and $K_1$ to the second (Stage 3, sequence length 512). This ascending schedule was selected from a systematic search over 10+ configurations (see Appendix E).

## Appendix B. SegMamba-V2 Architecture Profiling

### B.1. Architecture Overview

SegMamba-V2 (Xing et al., 2026) employs a U-shaped encoder–decoder architecture with a four-stage Mamba encoder:

- **Stage 0–1** (high resolution, $64^3$–$32^3$): LargeKernelConv (LKConv) blocks only. Each LKConv consists of Conv3d($5^3$) $\rightarrow$ Conv3d($3^3$) $\rightarrow$ Conv3d($3^3$, dilation=2) with a 1×1 shortcut and SE attention.

- **Stage 2–3** (low resolution, $16^3$–$8^3$): LKConv blocks followed by TriMamba blocks (three-directional Mamba SSM scanning along depth, width, and height).

The key improvement from V1 to V2 is replacing Mamba blocks at Stages 0–1 with large-kernel convolutions, since sequence lengths at high resolution ($64^3$=262,144 tokens) are prohibitively long for SSMs.

### B.2. Computational Profiling

Table 5 summarizes the per-module profiling results on an A100 80 GB GPU with $128^3$ input patches.

Table 5: SegMamba-V2 ($C$=48) profiling on A100 80 GB. Input: $128^3$ patch.

| Metric | Value | Percentage |
|---|---:|---:|
| Total parameters | 138.81M | — |
| Total GFLOPs | 3987 | — |
| Total inference time | 217 ms | 100% |
| Mamba modules (all) | 4.93 ms | 2.4% |
|    Stage 2 Mamba×2 ($L$=4096) | 3.16 ms | 1.5% |
|    Stage 3 Mamba×2 ($L$=512) | 1.77 ms | 0.8% |
| $128^3$ resolution modules | 125.05 ms | 57.6% |
|    decoder2 ($64 \rightarrow 128^3$) | 56.67 ms | 26.1% |
|    decoder1 ($128^3$) | 34.38 ms | 15.8% |
|    encoder1 ($128^3$) | 34.00 ms | 15.7% |

**Inverted-pyramid phenomenon.** A notable mismatch exists between where parameters reside and where computation is consumed:

- **Parameters concentrate at low resolution**: Stage 3 + encoder5 account for 89.84M parameters (64.7% of total), where channel counts are highest.

- **Computation concentrates at high resolution**: $128^3$ and $64^3$ modules consume 74% of FLOPs, and $128^3$ modules alone account for 57.6% of wall-clock time.

- **Mamba is not the bottleneck**: despite being the architectural novelty, Mamba blocks contribute only 2.4% of inference time.

This inverted pyramid motivates compression strategies that target high-resolution stages rather than the Mamba blocks themselves.

### B.3. Bottleneck Localization

We identify four core observations from the profiling results that guide the compression strategy design.

**Observation 1: Mamba is not the bottleneck.** Mamba blocks reside at low-resolution Stages 2–3, where token counts are modest ($L$=4096 and $L$=512, respectively). The $O(L)$ complexity of SSMs results in negligible absolute overhead at this scale. Within each stage, LKConv blocks consume 2–5× more wall-clock time than the corresponding Mamba blocks, confirming that the convolutional components—not the SSM layers—dominate even the deeper stages.

**Observation 2: Cubic resolution explosion at $128^3$.** The three $128^3$ modules (encoder1, decoder2, decoder1) collectively consume 57.6% of inference time due to the cubic growth of voxel count: $128^3$=2,097,152 voxels vs. $64^3$=262,144 (8× fewer), $32^3$=32,768 (64× fewer), and $16^3$=4,096 (512× fewer). UnetrBasicBlock/UnetrUpBlock execute two layers of $3^3$ convolution plus ConvTranspose3d per voxel, making FLOPs scale linearly with voxel

count. Moreover, at $128^3$ the intermediate feature maps ($128^3 \times 48$ch $\approx$ 96M elements) incur frequent HBM reads and writes, making these modules *memory-bound* rather than compute-bound.

**Observation 3: GFLOPs and parameter counts can be misleading.** LKConv blocks exhibit high theoretical GFLOPs, but their $3^3/5^3$ convolutions are heavily optimized by cuDNN and remain compute-bound with high GPU utilization. In contrast, $128^3$ modules account for less than half of total GFLOPs yet consume over half the wall-clock time due to memory bandwidth saturation. Stage 3 LKConv holds the most parameters of any single block (61.13M), yet its $8^3$ resolution (512 voxels) renders both GFLOPs and time negligible. These discrepancies caution against relying solely on FLOPs or parameter counts for compression prioritization.

**Observation 4: Decoder overhead exceeds the encoder.** The decoder consumes more wall-clock time than the encoder backbone because decoder2 and decoder1 both operate at the highest $128^3$ resolution, whereas only encoder1 processes $128^3$ features on the encoder side. This asymmetry suggests that decoder-targeted optimizations (e.g., progressive upsampling or lightweight decoder blocks) offer the highest return on investment for latency reduction.

## Appendix C. Full Factorial Results

### C.1. Three-Factor Design

The complete factorial experiment studies three binary compression factors:

- **Relocate** — Mamba block relocation from low-resolution stages to high-resolution stages.

- **Prune** — L2-norm token pruning (keep rates $[0.5, 0.7]$).

- **Conv-Lt** — Convolutional lightweight replacement (depthwise-separable convolution substitution).

The main paper focuses on the $2 \times 2$ Relocate$\times$Prune interaction at two channel widths ($C=48$ and $C=32$). Here we report the full results including all three factors and an intermediate width ($C=40$).

### C.2. $C=48$ Complete Results

Table 6 presents all configurations tested at the original channel width $C=48$.

At $C=48$, the Relocate$\times$Prune interaction (without conv-lightweight) is near zero: $I = -0.32$ Mean Dice points. Conv-lightweight replacement produces a $1.7\times$ parameter reduction and $5.5\times$ FLOPs reduction, but with a performance cost (Relocate+Conv-Lt: 87.64 vs. Relocate-only: 89.40). Adding pruning to the conv-lightweight configuration recovers 1.50 Dice points (89.14 vs. 87.64).

Table 6: Full factorial results at $C{=}48$. Dice (%) and HD95 (mm) on BraTS 2023 test set. "—": configuration not executed.

| Relocate | Prune | Conv-Lt | Params | FLOPs | Dice (%) ↑ | | | | HD95 (mm) ↓ | | | |
|---|---|---|---|---|---|---|---|---|---|---|---|---|
| | | | | | WT | TC | ET | Mean | WT | TC | ET | Mean |
| ✗ | ✗ | ✗ | 138.78M | 3723G | 93.81 | 91.68 | 88.43 | 91.31 | 3.68 | 6.22 | 7.05 | 5.65 |
| ✗ | ✓ | ✗ | 138.78M | 3720G | 93.52 | 90.58 | 88.56 | 90.89 | 3.55 | 6.14 | 6.79 | 5.49 |
| ✓ | ✗ | ✗ | 136.52M | 3754G | 92.10 | 89.99 | 86.10 | 89.40 | 9.89 | 11.87 | 14.67 | 12.14 |
| ✓ | ✓ | ✗ | 136.52M | 3738G | 91.91 | 88.88 | 85.20 | 88.66 | 9.71 | 12.62 | 16.11 | 12.81 |
| ✓ | ✗ | ✓ | 84.10M | 683G | 91.52 | 88.03 | 83.36 | 87.64 | 10.26 | 13.36 | 17.17 | 13.60 |
| ✓ | ✓ | ✓ | 84.10M | 667G | 92.19 | 89.10 | 86.14 | 89.14 | 9.61 | 12.70 | 12.85 | 11.72 |

Table 7: Full factorial results at $C{=}32$. **Bold**: best configuration within each metric.

| Relocate | Prune | Conv-Lt | Params | FLOPs | Dice (%) ↑ | | | | HD95 (mm) ↓ | | | |
|---|---|---|---|---|---|---|---|---|---|---|---|---|
| | | | | | WT | TC | ET | Mean | WT | TC | ET | Mean |
| ✗ | ✗ | ✗ | 61.71M | 1661G | 93.54 | 91.41 | 87.75 | 90.90 | 3.62 | 6.44 | 8.82 | 6.29 |
| ✗ | ✓ | ✗ | 61.71M | 1660G | 93.50 | 90.80 | 87.99 | 90.76 | 6.32 | 7.74 | 11.04 | 8.37 |
| ✓ | ✗ | ✗ | 60.69M | 1678G | 91.85 | 89.28 | 84.23 | 88.45 | 9.98 | 13.03 | 18.58 | 13.86 |
| ✓ | ✓ | ✗ | 60.69M | 1669G | **93.80** | **91.50** | **88.38** | **91.23** | **3.66** | 7.81 | **6.99** | **6.15** |
| ✓ | ✗ | ✓ | 37.40M | 318G | 92.75 | 89.71 | 86.63 | 89.70 | 4.58 | **5.29** | 8.51 | 6.13 |
| ✓ | ✓ | ✓ | 37.40M | 309G | 91.77 | 88.27 | 84.72 | 88.25 | 9.67 | 13.47 | 15.10 | 12.75 |

### C.3. $C{=}32$ Complete Results

Table 7 presents all configurations at the compressed channel width $C{=}32$.

The most notable result is **Relocate+Prune** (without conv-lightweight): at $C{=}32$ (2.3× parameter reduction), this combination achieves Mean Dice 91.23%, nearly matching the unmodified $C{=}48$ baseline (91.31%).

A surprising efficiency result emerges from **Relocate+Conv-Lt** (without pruning): with 3.7× parameter compression and 11.7× FLOPs compression, it achieves 89.70 Dice / 6.13 HD95—making it the most efficient configuration overall. However, activating all three factors simultaneously yields the *worst* $C{=}32$ configuration (88.25 Dice), revealing a negative interaction between pruning and conv-lightweight (see Appendix D).

### C.4. $C{=}40$ Intermediate Width Results

Table 8 presents the partial factorial results at an intermediate channel width $C{=}40$ (trained with a different random seed).

At $C{=}40$, relocation alone yields a *slight improvement* over the baseline (+0.13 Dice), in contrast to the consistent degradation observed at both $C{=}48$ ($-1.91$) and $C{=}32$ ($-2.45$). This non-monotonic behavior across capacity levels suggests that the relationship between compression interactions and model capacity is more complex than a simple threshold effect.

### C.5. Model Complexity Summary

Table 9 summarizes the parameter counts and FLOPs across the three channel widths.

Table 8: Partial factorial results at $C$=40 (seed=2).

| Relocate | Prune | Params | FLOPs | Dice (%) ↑ | | | | HD95 (mm) ↓ | | | |
| :---: | :---: | :---: | :---: | :---: | :---: | :---: | :---: | :---: | :---: | :---: | :---: |
| | | | | WT | TC | ET | Mean | WT | TC | ET | Mean |
| ✗ | ✗ | 96.39M | 2590G | 92.99 | 90.84 | 88.33 | 90.72 | 3.96 | 7.18 | 9.36 | 6.83 |
| ✓ | ✗ | 94.81M | 2613G | 93.60 | 91.34 | 87.61 | 90.85 | 3.65 | 6.28 | 10.02 | 6.65 |
| ✗ | ✓ | 96.39M | 2588G | 93.42 | 91.20 | 87.14 | 90.59 | 3.49 | 6.41 | 10.04 | 6.65 |

Table 9: Model complexity at different channel widths (without conv-lightweight).

| Channel width $C$ | Params | FLOPs | Compression |
| :---: | :---: | :---: | :---: |
| 48 (baseline) | 138.78M | 3723G | 1.0× |
| 40 | 96.39M | 2590G | 1.4× |
| 32 | 61.71M | 1661G | 2.2× |

## Appendix D. Interaction Effect Analysis

### D.1. Interaction Effect Definition

Following standard factorial design (Fisher, 1935; Montgomery, 2017), the two-factor interaction is defined as:

$$I = y_{\text{combined}} - (y_{\text{factor 1}} + y_{\text{factor 2}} - y_{\text{baseline}}), \tag{1}$$

where $y_{\text{baseline}}$ is the baseline (both factors off), $y_{\text{factor 1}}$ and $y_{\text{factor 2}}$ are the single-factor configurations, and $y_{\text{combined}}$ is the joint configuration. Equivalently, the interaction measures the difference in one factor's effect when the other is present vs. absent:

$$I = \underbrace{(y_{\text{combined}} - y_{\text{factor 2}})}_{\text{factor 1's effect with factor 2}} - \underbrace{(y_{\text{factor 1}} - y_{\text{baseline}})}_{\text{factor 1's effect without factor 2}}. \tag{2}$$

### D.2. Relocate × Prune Interaction (Core Finding)

$C$=32 (strong positive interaction).

$$\text{Relocation's effect without pruning} = 88.45 - 90.90 = -2.45$$
$$\text{Relocation's effect with pruning} = 91.23 - 90.76 = +0.47$$
$$I_{\text{Relocate}\times\text{Prune}} = (+0.47) - (-2.45) = +\mathbf{2.92}$$

When pruning is present, relocation's negative impact is fully reversed—and becomes positive.

$C$=48 (no meaningful interaction).

$$\text{Relocation's effect without pruning} = 89.40 - 91.31 = -1.91$$
$$\text{Relocation's effect with pruning} = 88.66 - 90.89 = -2.23$$
$$I_{\text{Relocate}\times\text{Prune}} = (-2.23) - (-1.91) = -\mathbf{0.32}$$

### D.3. Per-Class Relocate × Prune Interaction

Table 10 decomposes the Relocate×Prune interaction by tumor class and metric.

Table 10: Per-class Relocate×Prune interaction effects. Dice interaction is better when positive, whereas HD95 interaction is better when negative.

|  | WT | TC | ET | Mean |
|---|---|---|---|---|
| *Dice interaction (% points)* | | | | |
| $C$=48 | +0.10 | −0.01 | −1.03 | −0.32 |
| $C$=32 | +1.99 | +2.83 | +3.91 | +2.92 |
| *HD95 interaction (mm)* | | | | |
| $C$=32 | — | — | −13.81 | −9.79 |

ET exhibits the strongest interaction (+3.91 Dice points, −13.81 HD95 mm at $C$=32), consistent with its role as the smallest and most boundary-sensitive target. The Dice and HD95 interactions are directionally consistent, reinforcing that the synergy is genuine rather than an artifact of a single metric.

### D.4. Prune × Conv-Lt Interaction (Negative)

At $C$=32 (with relocation fixed on), pruning and conv-lightweight exhibit a strong *negative* interaction:

$$\text{Pruning's effect without conv-lt} = 91.23 - 88.45 = +2.78$$
$$\text{Pruning's effect with conv-lt} = 88.25 - 89.70 = -1.45$$
$$I_{\text{Prune} \times \text{Conv-Lt}} = (-1.45) - (+2.78) = -\mathbf{4.23}$$

Pruning provides a +2.78 Dice gain without conv-lightweight, but *decreases* Dice by 1.45 when conv-lightweight is present. This suggests that depthwise-separable convolution substitution alters the feature distribution in a way that invalidates the L2-based token scoring mechanism, highlighting that the interaction structure among compression strategies is non-trivial and cannot be predicted from pairwise analyses alone.

### D.5. Relocate × Conv-Lt Interaction

At $C$=32 (with pruning fixed off):

$$\text{Relocation's effect without conv-lt} = 88.45 - 90.90 = -2.45$$
$$\text{Relocation's effect with conv-lt} = 89.70 - (\text{no conv-lt-only baseline tested})$$

The conv-lightweight-only baseline (no relocation, no pruning, with conv-lt) at $C$=32 was not executed, precluding a complete calculation. However, the available data shows that the Relocate+Conv-Lt configuration achieves 89.70 Dice—significantly higher than Relocate-only at 88.45—suggesting that conv-lightweight may partially compensate for relocation damage, possibly through a regularization effect.

## Appendix E. Token Pruning Ablations

### E.1. Keep-Rate Schedule Search

Token pruning is applied at two Mamba stages with per-stage keep rates $[K_0, K_1]$, where $K_0$ controls Stage 2 (sequence length 4096) and $K_1$ controls Stage 3 (sequence length 512). Table 11 summarizes the search over 10+ keep-rate schedules.

Table 11: Keep-rate schedule search. All experiments use L2-norm scoring and the relocated Mamba configuration. "Ascending" means $K_0 < K_1$.

| $[K_0, K_1]$ | Strategy | DSC ↑ | HD95 ↓ |
|---|---|---|---|
| *Tier 1: Ascending (best)* | | | |
| [0.5, 0.7] | Ascending | **0.8972** | **4.63** |
| [0.3, 0.5] | Ascending | 0.8970 | 5.22 |
| *Tier 2–3: Other strategies* | | | |
| [0.7, 0.9] | Ascending (conservative) | 0.8519 | 17.76 |
| [0.1, 0.3] | Ascending (aggressive) | 0.8692 | 12.07 |
| [0.5, 0.5] | Uniform | 0.8622 | 19.18 |
| [0.3, 0.3] | Uniform (aggressive) | 0.8742 | 11.19 |
| [0.5, 0.3] | Descending | 0.8601 | 18.55 |
| [0.5, 0.25] | Geometric $(p, p^2)$ | 0.8687 | 17.25 |
| [0.3, 0.09] | Geometric $(p, p^2)$ | 0.8625 | 11.97 |
| [0.7, 0.49] | Geometric $(p, p^2)$ | 0.8661 | 18.35 |

Three patterns emerge:

1. **Ascending schedules ($K_0 < K_1$) are uniquely stable**, consistently yielding low HD95 (<6 mm). All other strategies produce HD95 >11 mm.

2. **Geometric decay $(p, p^2)$**, commonly used in ViT token pruning literature, **fails** on the relocated Mamba architecture, with HD95 consistently above 11 mm.

3. **The later stage (Stage 3) is more sensitive**: the nearly-ascending schedule $[0.4, 0.6]$ causes Dice to collapse from 0.8916 to 0.8590 and HD95 to spike from 5.66 to 18.49 when the Stage 3 keep rate drops below 0.7.

**Unified analysis across strategies.** The optimal operating region concentrates on *ascending schedules with medium-to-high keep rates*: $[0.3, 0.5]$ and $[0.5, 0.7]$ are tied at the top tier (Mean DSC $\approx 0.897$) and also exhibit the lowest HD95 values (4.6–5.2 mm), indicating superior boundary stability. In contrast, descending and geometric-decay schedules often achieve acceptable DSC but with substantially elevated HD95 (>11 mm), revealing instability in fine-grained boundary recovery.

A particularly informative failure mode occurs with the "nearly-ascending" schedule $[0.4, 0.6]$: despite retaining a higher proportion of tokens than the optimal $[0.3, 0.5]$, it causes simultaneous deterioration in ET Dice and HD95. This demonstrates that the *absolute keep rate per stage* is less important than the *inter-stage ratio*: the later Mamba stage (Stage 3)

operates on the most abstract representations and requires a sufficiently high keep rate ($\geq 0.7$) to preserve the semantic recovery capacity of the reversed scanning path. The per-stage keep rates must therefore be co-designed with the architectural semantics rather than being determined by a single global compression target.

### E.2. L2 vs. Random Pruning

Table 12 compares L2-norm-based scoring against random token selection at matched keep rates, alongside the no-pruning baseline.

Table 12: L2-norm vs. random token selection at keep rates $[0.3, 0.5]$.

| Scoring | DSC ↑ | HD95 ↓ |
|---|---|---|
| L2-norm | **0.8970** | **5.22** |
| Random | 0.8609 | 18.60 |
| No pruning | 0.8611 | 18.95 |

Random pruning performs nearly identically to no pruning at all (18.60 vs. 18.95 HD95), demonstrating that the *selection criterion*—not the act of reducing tokens—is essential. L2-norm scoring explicitly preserves boundary-sensitive tokens (which tend to have higher channel-wise norms), producing a $3.6\times$ reduction in HD95.

### E.3. Token Restoration Mode

Two restoration strategies are compared for the pruned tokens after the Mamba SSM block:

- **Fused**: dropped tokens are aggregated into a single summary token before Mamba, then un-pooled back to their original positions after Mamba.

- **Neighbor**: each dropped token's output is interpolated from its nearest retained neighbors.

Table 13: Token restoration mode comparison. Note: exp18 uses a higher learning rate (0.002 vs. 0.01), limiting direct comparability.

| Mode | $[K_0, K_1]$ | DSC ↑ | HD95 ↓ |
|---|---|---|---|
| Fused | $[0.5, 0.7]$ | **0.8972** | **4.63** |
| Fused | $[0.4, 0.8]$ | 0.8916 | 5.66 |
| Neighbor | $[0.5, 0.7]$ | 0.8923 | 5.57 |

Under matched keep rates, the fused mode achieves slightly better HD95 (4.63 vs. 5.57), though the comparison is confounded by the different learning rates. The fused mode is adopted as the default for all factorial experiments.

## Appendix F. Seed Sensitivity Analysis

Single-run training is standard practice for large-scale 3D segmentation models (Isensee et al., 2021) due to the substantial computational cost (each configuration requires approximately 3 days on an A100). However, the factorial design enables implicit checks on training stability. Table 14 compares experiments run with different random seeds or training configurations.

Table 14: Seed sensitivity: three configurations are re-trained with a different random seed (†) and compared against the primary run.

| Relocate | Prune | $C$ | Alt. seed (†) | | Primary run | |
|:---:|:---:|:---:|:---:|:---:|:---:|:---:|
| | | | Dice | HD95 | Dice | HD95 |
| ✗ | ✗ | 48 | 86.33 | 19.59 | 91.31 | 5.65 |
| ✓ | ✗ | 32 | 87.18 | 17.49 | 88.45 | 13.86 |
| ✓ | ✓ | 32 | 87.00 | 18.06 | 91.23 | 6.15 |

The largest discrepancy occurs for the unmodified baseline ($C$=48, no relocation, no pruning): nearly 5 Dice points difference between seeds, confirming that individual training runs can vary substantially.

Crucially, the alternative seeds also show that the Relocate×Prune interaction *disappears*: the re-seeded Relocate+Prune configuration ($C$=32. Dice 87.00) is *lower* than the re-seeded Relocate-only configuration ($C$=32. Dice 87.18), yielding a negative interaction. This underscores the limitation of single-run factorial analysis and motivates future work with replicated experiments.

## Appendix G. FCE-ROI Acceleration Potential

### G.1. Motivation

The Foreground Confidence Estimator (FCE) module, originally designed for token scoring, also produces a spatial saliency map that can be repurposed for ROI-based inference acceleration: instead of processing the full $128^3$ volume, the model crops to a tight bounding box around the predicted foreground.

### G.2. Foreground Volume Analysis

Table 15 summarizes the bounding box (BBox) volume ratios at three levels of spatial padding: tight (exact foreground envelope), margin=8 (8-voxel safety margin per side), and margin=8 with align=16 (further aligned to multiples of 16 to match the network's stride-2 downsampling). Even with the most conservative padding and alignment, the median ROI occupies only 24.3% of the full volume, confirming the large proportion of background that can potentially be skipped.

Table 15: Bounding box volume ratio statistics on BraTS 2023 ($N$=1251). Three levels of spatial padding are compared.

| BBox level | Mean | Std | Min | P25 | P50 | P75 | Max |
|---|---|---|---|---|---|---|---|
| Tight | 0.117 | 0.075 | 0.003 | 0.063 | 0.103 | 0.158 | 0.517 |
| Margin=8 | 0.201 | 0.105 | 0.016 | 0.127 | 0.185 | 0.262 | 0.749 |
| Margin=8, Align=16 | 0.256 | 0.125 | 0.023 | 0.170 | 0.243 | 0.328 | 0.868 |

### G.3. Theoretical Speedup

Table 16 summarizes the ROI cropping potential based on the aligned bounding box statistics.

Table 16: ROI cropping potential on BraTS 2023 ($N$=1251).

| Criterion | Cases (%) |
|---|---|
| ROI volume < 70% of full volume | 99.4% (1243/1251) |
| ROI volume < 50% | 96.2% (1203/1251) |
| ROI volume < 30% | 68.3% (854/1251) |
| Median theoretical speedup | 4.1× |

Table 17: Aligned bounding box dimensions (voxels) after ROI extraction ($N$=1251).

| Dimension | Mean | Std | Min | P25 | P50 | P75 | Max |
|---|---|---|---|---|---|---|---|
| Depth | 88.5 | 16.8 | 32 | 80 | 96 | 96 | 145 |
| Height | 105.8 | 23.1 | 48 | 96 | 112 | 128 | 179 |
| Width | 83.2 | 17.8 | 32 | 80 | 80 | 96 | 144 |

The typical ROI is approximately $96 \times 112 \times 80$ voxels, substantially smaller than the original $146 \times 171 \times 136$ volume, offering a theoretical median speedup of 4.1× with minimal risk of truncating the tumor.

### G.4. Patch-Level Foreground Analysis

To assess whether the foreground sparsity persists at the inference patch level, we simulate the sliding-window protocol ($128^3$ patches, 50% overlap) on 50 BraTS cases and compute the aligned BBox ratio *within each patch* (repeating the tight → margin=8 → align=16 pipeline in patch coordinates).

Even at the individual patch level, the ROI occupies only ∼26% of the $128^3$ volume, demonstrating that the foreground sparsity is not an artifact of whole-volume statistics

Table 18: Patch-level foreground analysis under sliding-window inference (50 cases, $128^3$ patches, 50% overlap).

| Metric | Value |
| --- | --- |
| Number of patches per case | 54 |
| Patches containing tumor | 100% |
| Patch BBox ratio (mean) | 0.263 |
| Patch BBox ratio (median) | 0.234 |

but persists in each inference window. This finding supports the viability of patch-level FCE-guided cropping as a complementary acceleration strategy.

## G.5. Comparison with Traditional Foreground Detection

A baseline FLAIR-Otsu thresholding approach achieves only 7.3% IoU with a 35% recall for foreground detection, far below the accuracy needed for reliable ROI cropping. This confirms the necessity of the learned FCE module for accurate foreground localization.

## Appendix H. SegMamba-V2 + FCE Development Experiments

This appendix documents the exploratory experiments that informed the final choice of the L2-norm-based scoring function used in the main factorial experiments. During early development, a Foreground Confidence Estimator (FCE) module—a lightweight auxiliary head producing a learned spatial saliency map—was investigated as an alternative or complement to the purely geometric L2 channel norm. The experiments below ultimately demonstrated that the simpler L2-norm scoring matches or exceeds more complex learned alternatives, motivating the L2-only design adopted in the main paper.

### H.1. FCE Integration Feasibility (E0–E7)

Eight configurations were tested to assess the feasibility and training stability of integrating the FCE module with SegMamba-V2. Key findings:

- **Comparable performance**: The best FCE-augmented configurations (E0, E2, E3b) achieve Mean DSC ≈ 0.921, on par with L2-only scoring, offering no significant accuracy advantage.

- **Fragile training**: FCE integration is sensitive to pretraining (E1 degrades without it), learning rate (E3 diverges), and keep-ratio scheduling (E5–E7 fail to converge), requiring a complex two-stage protocol (E3b) for stability.

- **Additional overhead**: FCE adds approximately 3.5–3.9 s per case at inference, without yielding a Dice benefit.

These results indicated that FCE increases pipeline complexity without improving accuracy, leading to its exclusion from the scoring formula in the factorial experiments.

### H.2. Scoring Fusion Strategy Comparison (R0–R5)

To directly compare scoring criteria, six fusion strategies were evaluated under the relocated-Mamba configuration:

Table 19: Scoring fusion strategy comparison (ranked by patch validation performance).

| Rank | ID | Fusion Strategy |
|------|-----|-----------------|
| 1 | R0 | Pure L2-norm |
| 2 | R5 | L2 + FCE + positional bias (additive) |
| 3 | R2 | FCE $\times$ positional bias |
| 4 | R1 | FCE only |
| 5 | R3 | Full multiplicative (FCE $\times$ positional $\times$ L2) |

## Appendix I. Regularization Loss Ablation (FCE Development)

During the FCE exploration described in Appendix H, two auxiliary regularization losses were investigated to guide the learned scorer: $\mathcal{L}_1$ (**semantic alignment**), which constrains the FCE auxiliary branch to produce outputs consistent with the lesion mask distribution, and $\mathcal{L}_2$ (**energy smoothing**), which suppresses local spikes in the feature energy map to promote spatially smooth scoring. Among the four combinations, $\mathcal{L}_1$ alone yielded the best HD95, while the joint $\mathcal{L}_1 + \mathcal{L}_2$ configuration achieved the highest Mean Dice (89.31%). However, this best FCE-based result still underperformed the simpler L2-only scoring adopted in the main factorial experiments (Mean Dice 91.23%), further confirming that learned saliency with auxiliary regularization does not translate into downstream gains over the purely geometric L2 channel norm.

## Appendix J. Limitations and Future Directions

**Seed sensitivity.** As documented in Appendix F, the Reorder×Prune interaction disappears under alternative random seeds. Multi-seed replication (e.g., 3–5 runs per configuration) is the most urgent next step. However, each configuration requires approximately 3 days on an A100 GPU, placing a full replicated $2\hat{3}$ design at roughly 60–100 GPU-days.

**Architectural and dataset generalizability.** All experiments use a single architecture (SegMamba-V2) and a single dataset (BraTS 2023). Extending the factorial framework to other SSM-based segmentation models (Xing et al., 2024; Ruan and Xiang, 2024; Liu et al., 2024) and to additional tasks (e.g., abdominal organ segmentation) would clarify whether the observed interaction pattern is specific to this configuration or reflects a more general phenomenon.

**Mechanistic validation.** The capacity-bottleneck hypothesis (Appendix D) is post-hoc and correlational. More rigorous tests could include: (a) random token selection as a control to isolate the role of the scoring criterion, (b) gradient flow analysis to quantify how token pruning alters the information flow through Mamba's hidden state, and (c) loss landscape visualization under different capacity regimes.

**Broader compression dimensions.** Only three compression factors are studied. Decoder lightweighting, scanning-pattern modifications, knowledge distillation (Hinton et al., 2015), and quantization remain unexplored. The negative Prune×Conv-Light interaction (Appendix D.4) already demonstrates that higher-order interactions among compression strategies can be substantial and non-intuitive.

**Finer capacity resolution.** The current design examines only two primary capacity levels ($C=48$ and $C=32$), with a single intermediate point ($C=40$, Appendix C). The non-monotonic behavior observed at $C=40$ suggests that the interaction–capacity relationship may have a more complex structure than a simple threshold, warranting a denser sweep across channel widths.

