# OpenReview forum: "Individually Harmful, Jointly Beneficial: Compression Strategy Interactions in Brain Tumor Segmentation"
_MIDL.io/2026/Short_Papers — MIDL 2026 - Short Papers Poster_

### Official Review · Reviewer_EECV · 2026-04-24
**Good paper but not suitable for short MIDL paper**

**Rating:** 3
**Confidence:** 4

**Review:**

The paper is an experimental study of how different elements of network architecture design affect its final performance. It is in line with the MIDL philosophy. However, considering the number of experiments done and the manner in which they are presented (many things are in a 12-page appendix), I am recommending rejection of the paper. I would encourage the authors to submit it to the conference as a pull paper in the future.

**Summary:**

The paper presents two contrasting findings related to the design of MAMBA architecture for Brain Tumour Segmentation. Through a set of thorough experiments, authors show that relocation of TriMamba blocks to high resolution, and L2 pruning alone leads to lower performance; however, when combined together, they recover the original model performance.

**Strengths:**

* Thorough experimental design.
* This type of paper is rare, which tries to understand the effect of different LEGO blocks on the network architecture. This is commendable.
* Good Limitation and Self-Critique section.

**Weaknesses:**

* A lot of the details are in the appendix of the paper. It makes it difficult to understand. This is particularly important considering the short nature of the MIDL short paper track.
* As mentioned by the authors, it is necessary to investigate if the proposed effect holds true across different runs and across different datasets.

**Justification Of Rating:**

The paper is good and within the line of MIDL philosophy. However, considering the length of the appendix and its necessity to understand all the different parts of the proposed experimental design, I recommend a borderline rating.

---

### Decision · Program_Chairs · 2026-05-08

Accept (Poster)